# OpenReview forum: "AGA: Attention-Guided Jailbreak Attacks on Large Reasoning Models"
_ICLR.cc/2026/Conference — ICLR 2026 Conference Withdrawn Submission_

### Official Review · Reviewer_Gd2a · 2025-10-19

**Soundness:** 2
**Presentation:** 3
**Contribution:** 3
**Rating:** 2
**Confidence:** 4

**Summary:**

This paper investigates why jailbreak attacks on Large Reasoning Models (LRMs) are successful, finding that effective attacks divert the model's attention away from harmful keywords in the prompt. Based on this insight, the authors propose AGA (Attention-Guided Jailbreak Attack), a novel method that iteratively refines prompts by using the LRM's intermediate reasoning steps to guide its focus. This attention-guided process helps balance the prompt's malicious intent with the need to remain unnoticed. Experiments conducted on five open-source and closed-source models show that AGA achieves a high attack success rate and surpasses existing methods in stealthiness, efficiency, and transferability

**Strengths:**

1. The finding that reducing attention to harmful words enables successful jailbreaks is interesting. (Although may need more justification as W1)
2. Building on its analytical findings, the paper introduces AGA (Attention-Guided Jailbreak Attack). AGA uses an LRM attacker to iteratively refine prompts. This refinement is explicitly guided by the model's intermediate reasoning steps and the attention scores on harmful keywords, creating a clear optimization objective to balance stealth and effectiveness.
3. The paper validates AGA through extensive experiments on five different open-source and closed-source LRMs (including Qwen3 models, DeepSeek-R1, O4-mini, and Gemini-2.5-Flash) across three distinct datasets, in terms of effectiveness, efficiency (requires few iterations to succeed), and stealthiness.

**Weaknesses:**

1. For the finding (2) in line 185, it is not clear whether the lower attention proportion to the harmful keywords is due to the lower proportion of keywords in all the tokens in the prompt or each harmful keyword receives a lower attention score. From the description of finding (3) "where harmful concepts are discussed actively", it seems the higher / lower proportion is attributed to the proportion of keywords in all the tokens in the prompt or the reasoning content?

2. It is not clear whether the nanoGCG in Sec. 3.3 only uses the custom loss function or the combination of the custom loss function and the original loss function.

3. There is no justification for using Llama-3.1-70B as a safety judge, as close-source models like GPT4-o are more common in prior works. The justification of the final safety judge result, like a human agreement study, is also missing.

4. State-of-the-art jailbreak attacking methods like AutoDAN-Turbo, Adaptive Attacks, X-Teaming are not compared. The compared baselines are relatively old.

5. The method requires white-box access to the LRMs, limiting its applicability to black-box models.

[1] AutoDAN-Turbo: A Lifelong Agent for Strategy Self-Exploration to Jailbreak LLMs
[2] Jailbreaking Leading Safety-Aligned LLMs with Simple Adaptive Attacks
[3] X-Teaming: Multi-Turn Jailbreaks and Defenses with Adaptive Multi-Agents

**Questions:**

1. Can the authors provide a citation for this statement? "Qwen3-8B exhibits behavior patterns and security vulnerabilities that are highly consistent with those observed in other leading LRMs".
2. It is not clear how the Qwen3-8B-abliterated version is achieved.

---

### Official Review · Reviewer_tXbC · 2025-10-23

**Soundness:** 2
**Presentation:** 2
**Contribution:** 3
**Rating:** 4
**Confidence:** 4

**Summary:**

This paper makes the following contributions:
- It analyzes both successful and failed jailbreak prompts, finding that in successful attacks, the target model’s attention tends to shift away from harmful keywords in the prompts.
- Building on this observation, the paper proposes AGA, an attention-guided jailbreak method that achieves high effectiveness and efficiency across various LRMs, outperforming SOTA attackers.

**Strengths:**

- The analysis of why jailbreaks on LRMs can be successful is useful.

- The proposed method AGA is effective and efficient.

**Weaknesses:**

Overall, the paper would benefit from improved clarity in writing and greater rigor in the experimental design and evaluation.

1. Line 46 mentions “from a theoretical perspective, current approaches lack a comprehensive analysis of their effectiveness”. However, this paper itself also does not provide a theoretical analysis, which makes the critique feel somewhat inconsistent.

2. In Figure 1:
- The definition of attention distribution/proportion is unclear.
- In the manual evaluation, how do you define and decide the harmful keywords?

3. In Figure 2:
- In the manual evaluation, how do you define and decide the harmful keywords and sensitive terms?
- How many annotators participated in this evaluation? If more than one, statistical uncertainty should be reported. If only one, the evaluation appears less reliable, as the definitions of harmful words and sensitive terms are highly subjective.
- If an auxiliary LLM were used to annotate harmful keywords and sensitive terms, would the conclusions in Section 3.2 remain the same?

4. What is the mathematical definition of the custom loss used in nanGCG’? How do you define the harmful keywords for this loss?

5. Several key terms are never explicitly defined, though they are indirectly introduced later in the paper. Their precise definitions should be provided earlier (e.g., in Algorithm 1) for clarity:
- Template/Template pool
- Extract
- ReduceAttention
- StrengthenPrompt.

6. In Section 4.3, how do you obtain the predefined harmful words dictionary D?

7. How do you instruct the attack model, judge model, and harmful phrase extractor? The paper does not provide sufficient details.

8. Line 342, what is the ‘predefined pool’ for the prompt initialization?

9. Line 361. How do you instruct the attacker model to “analyze safety concerns and revise the prompt accordingly”?

10. Line 371, how do you instruct the attacker model to strengthen the prompt?

11. Across all results, statistical uncertainty is not presented. Including such measures would strengthen the robustness and reliability of the reported findings.

**Questions:**

Please check **Weaknesses** for my questions.

**Details Of Ethics Concerns:**

No.

---

### Official Review · Reviewer_fuLm · 2025-10-27

**Soundness:** 3
**Presentation:** 3
**Contribution:** 2
**Rating:** 4
**Confidence:** 3

**Summary:**

The paper analyzes why jailbreaks succeed on Large Reasoning Models (LRMs) and finds a consistent pattern: successful attacks reduce attention to harmful words in the prompt while raising attention to them during the model’s reasoning. Building on this, it proposes AGA, an attention-guided loop that iteratively edits prompts.
Across three datasets and multiple LRMs, AGA attains the best average ASR-R/ASR-T, competitive perplexity (stealth), and very low average successful turns.

**Strengths:**

- Empirical insight to method design is clearly described.
- Strong results across metrics. AGA attains best average ASR-R/ASR-T with low PPL, and very low AST.
- Black-box transferability. Prompts crafted on Qwen3-8B reach about 50% ASR-R on o4-mini and Gemini-2.5-Flash.

**Weaknesses:**

- The method heavily relies on attention scores from the target model. When such scores are unavailable, as is common for closed-source or strictly black-box LRMs, the approach cannot effectively guide prompt updates, leading to degraded performance and limited real-world applicability.

- The paper's main empirical finding that successful attacks draw less attention to harmful words in the prompt and more attention to them during intermediate reasoning, is intuitive and aligns with existing understanding of attention redistribution. While it supports the method’s motivation, it offers limited new conceptual insight beyond confirming an expected pattern.

**Questions:**

- What is the underlying reason or necessity for pairing every target LRM with its unaligned counterpart as the attacker, instead of using a universal model as attacker?
- When the target model immediately rejects the prompt (without generating reasoning steps), AGA simply restarts with a template re-initialization. Is there a more principled or effective way to handle such Immediate Refusals beyond brute-force regeneration?

---

### Official Review · Reviewer_eZzt · 2025-10-30

**Soundness:** 3
**Presentation:** 3
**Contribution:** 3
**Rating:** 4
**Confidence:** 4

**Summary:**

This paper introduces AGA, an attention-guided jailbreak framework for Large Reasoning Models (LRMs). It is founded on the key insight that successful attacks divert the model's attention away from harmful keywords and towards its internal reasoning process. AGA operationalizes this finding by using the model's attention scores as a direct feedback signal to iteratively and efficiently refine prompts, achieving high success rates with strong stealth and efficiency.

**Strengths:**

- Novel and Well-Designed Attack Algorithm: The AGA framework is highly novel and elegantly designed. It innovatively uses an internal model state as an explicit signal to guide a well-defined feedback loop. The algorithm 1, which decides whether to "Reduce Attention" or "Strengthen Prompt" based on an attention threshold and a harmfulness score, is both intuitive and highly effective. Compared to many baseline methods that rely on black-box optimization or trial-and-error, AGA's optimization process is remarkably goal-oriented, which directly translates to its exceptional efficiency.
- Comprehensive and Rigorous Experimental Evaluation: The experimental design is thorough and robust. The authors evaluate AGA on multiple open-source models and demonstrate its generalization by transferring the generated prompts to state-of-the-art closed-source models. The choice of metrics is comprehensive, covering the attack's effectiveness, stealthiness, and efficiency. The introduction of ASR-T is a particularly insightful addition that is tailored to the unique characteristics of LRMs and highlights the authors' deep understanding of the problem space.

**Weaknesses:**

- Limited Novelty and Weak Theoretical Grounding: Although the paper proposes AGA as an “attention-guided jailbreak” framework, its conceptual contribution is incremental over prior attention-based jailbreak studies. The main idea (reducing the model’s attention on harmful tokens to evade safety detection) largely extends existing attention-shifting observations without introducing new theoretical insight or formal justification.
- Potential for Confounding Variables in Causal Claims from Attention Analysis: The preliminary study convincingly demonstrates a strong correlation between attention shifting and jailbreak success, and the paper treats this as a causal link to build its algorithm. However, confounding variables may exist. For instance, successful prompts (like P2 in Figure 2) are often longer, more linguistically complex, and use more nuanced phrasing. These characteristics alone could lead to a successful jailbreak, making the reduced attention on harmful words a symptom of this complexity rather than the root cause of success.

**Questions:**

Seen in weakness

---

### Note · Authors · 2025-11-21

I have read and agree with the venue's withdrawal policy on behalf of myself and my co-authors.